# Gradient Descent and the Power Method:
# Exploiting their connection to find the leftmost eigen-pair and escape saddle points

## Abstract

Applying Gradient Descent with fixed Momentum (GDM) and a fixed step size to minimize a (possibly nonconvex) quadratic function is equivalent to running the Power Method with fixed Momentum (PMM) on the gradients. Thus, valuable eigen-information is available via GDM. A new algorithm called Gradient Descent with a Kick (GD-Kick) is presented, which exploits the 'free' eigen-information available from the GDM-PMM connection, and occasionally takes a locally adaptive, long step. Numerical experiments show the advantages of GD-Kick compared with vanilla GD, particularly near saddle points.

## 1 Introduction

This work considers optimization problems of the form

$$\min_{x \in \mathbb{R}^n} f(x) \quad \text{where} \quad f(x) = \tfrac{1}{2}x^T A x - b^T x, \tag{1}$$

where $A \in \mathbb{R}^{n \times n}$ is symmetric, and $x, b \in \mathbb{R}^n$. If $A$ is Positive Definite (PD), then equation 1 is a strongly convex optimization problem. However, if $A$ is indefinite then equation 1 is nonconvex. Quadratic optimization problems arise in a plethora of situations, either directly, or perhaps as a subproblem within an iterative method. For example, many machine learning algorithms require a quadratic model to be minimized to determine an appropriate search direction within a trust region algorithm Nocedal & Wright (2006); Erway et al. (2020), or within a Newton CG algorithm Nocedal & Wright (2006); Royer et al. (2020). Moreover, nonconvex functions that are not quadratic are often well approximated by a quadratic in the neighbourhood of a local minimizer or saddle point. Hence, the study of quadratic optimization problems is important.

One of the most well known algorithms in optimization is Gradient Descent (GD). Its popularity abounds because it is simple to use and understand, is widely applicable, and scales to high dimensional problems. Much can be said about the behaviour of GD on convex optimization problems, including characterisations of the convergence guarantees, and rates of convergence. However, it is also known to converge slowly in practice, particularly on ill-conditioned problems.

Recently, partially motivated by the rise of applications in machine learning, attention has focused on the performance of GD on nonconvex problems. It is known that GD applied to nonconvex functions can find a stationary point in polynomial time Nesterov (2004), and that GD almost always escapes saddle points asymptotically Lee et al. (2016), but there are other works Du et al. (2017); Paternain et al. (2019) that present nonpathalogical examples suggesting that GD can take exponential time to escape saddle points. The work Du et al. (2017) asks: "Does randomly initialized GD generally escape saddle points in polynomial time?" and then presents examples demonstrating "a strong *negative* answer to this question".

The current work revisits the examples presented in Du et al. (2017); Paternain et al. (2019) and investigates whether the 'free' eigenvalue information available by running GD can be utilised to aid the performance of GD near saddle points. In doing so, a connection between GD with a fixed step size and the PM, both with and without fixed momentum, is established. In particular, GD with a fixed step size (with momentum)

is implicitly running the PM (with momentum) on the successive gradients. Further, it is shown that this connection can be exploited to provide an estimate of the leftmost eigenvalue-eigenvector pair of the Hessian at the cost of a couple of additional inner products. An algorithm called Gradient Descent with a Kick (GD-Kick) is presented, which is GD but periodically, local curvature information is incorporated into the algorithm in the form of an adaptive step size (a long step or 'kick'), with the intention of gaining a larger reduction in the function value at that iteration. The algorithm returns the minimizer of a quadratic function, and an approximate left eigen-pair.

## 1.1 Literature review

Gradient Descent is ubiquitous in optimization and machine learning. For convex optimization problems with a Lipschitz continuous gradient, GD converges at the (sublinear) rate $\mathcal{O}(1/k)$, while for strongly convex problems, GD exhibits linear converge at the rate $1 - 1/\kappa$, where $\kappa$ denotes the condition number. These rates of convergence are suboptimal for strongly convex problems Nesterov (2004), and much effort has been made developing novel GD based algorithms that converge faster.

One strategy that can help to improve the practical behaviour of GD is the heavy ball approach of Polyak Polyak (1964), where a 'momentum' term is added to the iteration update. While the iterates of GD are typically 'slow and steady', momentum helps to pull the iterates downhill more rapidly. Adding momentum can be beneficial, but it can be difficult to tune the momentum parameter, and the algorithm is not guaranteed to converge in general.

Nesterov was instrumental in the development of acceleration Nesterov (1983; 2005; 2013), and the accelerated gradient method is guaranteed to converge at the optimal linear rate $1 - 1/\sqrt{\kappa}$ for strongly convex functions. This algorithm works well in practice, but the strong convexity constant is often used in the algorithm, and it can be challenging to compute. Many variants of the accelerated gradient method exist, including Chen et al. (2017); Diakonikolas & Orecchia (2019); Drusvyatskiy et al. (2018); Ghadimi & Lan (2012); Jahani et al. (2021), which also provide a certificate of optimality.

For strongly convex quadratic optimization problems, Conjugate Gradients (CG) Hestenes & Stiefel (1952) is a successful algorithm, which is guaranteed to converge with rate $(\sqrt{\kappa} - 1)/(\sqrt{\kappa} + 1)$. Relevant to the current work is the connection between CG for solving problems of the form equation 1 when $A$ is positive definite, and the Lanczos method Lanczos (1950) for finding the extremal eigenvalues (and eigenvectors) of $A$. In particular, if CG is applied to equation 1, then approximations to the extremal eigenvalue(s) of $A$ can be recovered. Approximations to the corresponding eigenvectors are also available if the CG residual vectors are stored. (See also, for example, (Golub & Loan, 1996, p.528), (Demmel, 1997, p.307), (Saad, 2003, p.221) and Lanczos (1952); Meurant (2006).) Note that for CG, equation 1 must be strongly convex ($A$ must be PD). However, MINRES Paige & Saunders (1975) can be used even when $A$ is not PD, and the algorithm can also be used to generate eigen-information for $A$ via an analogous MINRES-Lanczos relationship. (See also (Golub & Loan, 1996, p.494).)

## 1.2 Contributions

The main contributions of this work are summarized now.

- *GDM–PMM Connection.* This work formalizes the connection between GD with a fixed step size and the Power Method (both with or without momentum). This relationship is analoguous to the connection between conjugate gradients and the Lanczos method.

- *Escaping saddle points.* Several examples from Du et al. (2017); Paternain et al. (2019), examining the behaviour of GD with a fixed step size near saddle points, are revisited. It is shown that an approximation to the leftmost eigen-pair is implicitly available, and strategies that use this information are suggested, providing potential ways of improving the practical performance of GD on these difficult nonconvex ('locally quadratic') problems.

- *Gradient Descent with a Kick.* New strategies are proposed to demonstrate how the eigen-information available via GD with a fixed step size might be used to improved the algorithm's behaviour. An

algorithm called GD-Kick is presented, which is motivated by the GD-PM connection, and preliminary numerical experiments show that it has better practical performance compared with vanilla GD.

### 1.3 Preliminaries and Notation

The following preliminaries and assumption are made throughout this work.

**Assumption 1.** *The matrix $A \in \mathbb{R}^{n \times n}$ is symmetric. The eigen-pairs of $A$ are denoted by $(\lambda_i, v_i)$ for $i = 1, \ldots, n$, with $\lambda_n \leq \cdots \leq \lambda_1$, and $|\lambda_n| \leq |\lambda_1|$.*

The gradient of $f$ is denoted as

$$g(x) = Ax - b. \tag{2}$$

(Usually the explicit dependence on $x$ is dropped, and the gradient is denoted by $g$). The gradient of $f$ is $L$-Lipschitz continuous, i.e., for all $x, y \in \mathbb{R}^n$ it holds that

$$\|\nabla f(x) - \nabla f(y)\|_2 \leq L \|x - y\|_2 \tag{3}$$

where for $f$ in equation 1, $L = \|A\|_2 = \lambda_1$.

If $A$ is symmetric and Positive Definite (PD) then equation 1 is $\mu$-strongly convex,
i.e., $f(x) + \nabla f(x)^T (y - x) + \frac{\mu}{2} \|y - x\| \leq f(y)$ for any $x, y \in \mathrm{dom}(f)$, and where $\mu = \lambda_n > 0$.

Throughout this work, the convention $Y^0 = I$, where $Y \in \mathbb{R}^{n \times n}$ and $I$ denotes the identity matrix, is adopted (i.e., a matrix raised to the power zero is the identity matrix). The vector $e_i \in \mathbb{R}^n$ denotes the $i$th column of the $n \times n$ identity matrix. Finally, $x^*$ denotes a minimizer of equation 1, while the set of minimizers of is denoted by $X^*$.

### 1.4 Outline

Section 2 establishes the connection between Gradient Descent with momentum and the Power Method with momentum. Section 3 revisits the examples in Du et al. (2017); Paternain et al. (2019) and shows that if eigenvalue information is taken into account, and the step size of GD is modified, then the practical behaviour of GD near saddle points may be better than previously suggested. Section **??** introduces the Gradient Descent with a Kick (GD-Kick) algorithm, which uses the approximation to the leftmost eigenvalue to give GD a 'kick' (a 'long' step) towards the solution, and numerical experiments are presented in Section 5 to demonstrate the practical behaviour of GD-Kick.

## 2 Gradient Descent and the Power Method

Given initial points $x^{(0)}, x^{(-1)} \in \mathbb{R}^n$ with $x^{(-1)} = x^{(0)}$, for $k \geq 0$, Gradient Descent with Momentum (GDM), with a fixed step size $\alpha$ and fixed momentum term $\beta$, is characterized by the iteration:

$$x^{(k+1)} = x^{(k)} - \alpha g^{(k)} + \beta(x^{(k)} - x^{(k-1)}), \tag{4}$$

where $g^{(k)} := g(x^{(k)})$. In words, to generate a new iterate $x^{(k+1)}$, a step is taken from the current point $x^{(k)}$ in the direction of the negative gradient of (fixed) size $\alpha$, and an adjustment term is added in the direction $x^{(k)} - x^{(k-1)}$ scaled by (fixed) momentum parameter $\beta$. Setting $\beta = 0$ recovers vanilla GD.

It is convenient to define the matrices:

$$H := (I - \alpha A), \tag{5}$$

and

$$\hat{H} := (1 + \beta)I - \alpha A \equiv H + \beta I, \tag{6}$$

and note that if $\beta = 0$ (no momentum) then $\hat{H} = H$. The following lemma shows how the gradient evolves when running GDM with fixed step-size $\alpha$, and fixed momentum parameter $\beta$.

**Lemma 1.** *Let Assumption 1 hold, let $\alpha, \beta \geq 0$ be fixed, and let $H$ and $\hat{H}$ be defined in (5) and (6), respectively. Given an initial point $x^{(0)} \in \mathbb{R}^n$, with $x^{(-1)} = x^{(0)}$, at any iteration $k \geq 0$ of Gradient Descent with Momentum applied to $f$ in (1),*

$$g^{(k+1)} = \hat{H}g^{(k)} - \beta g^{(k-1)}. \tag{7}$$

*Proof.* The gradient for equation 1 is

$$
\begin{aligned}
g^{(k+1)} \;&\overset{(2)}{=}\; Ax^{(k+1)} - b \\
&\overset{(4)}{=}\; A\left(x^{(k)} - \alpha g^{(k)} + \beta(x^{(k)} - x^{(k-1)})\right) - b \\
&=\; g^{(k)} - \alpha A g^{(k)} + \beta A(x^{(k)} - x^{(k-1)}) \\
&\overset{(5)}{=}\; Hg^{(k)} + \beta A(x^{(k)} - x^{(k-1)}) \\
&\overset{(2)}{=}\; Hg^{(k)} + \beta(g^{(k)} - g^{(k-1)}) \\
&\overset{(6)}{=}\; \hat{H}g^{(k)} - \beta g^{(k-1)}.
\end{aligned} \tag{8}
$$

$\square$

**Corollary 1.** *Let the conditions of Lemma 1 hold with $\beta = 0$. Then*

$$g^{(k+1)} = Hg^{(k)} = H^k g^{(0)}. \tag{9}$$

**Remark 1.** *Note that $\hat{H}$ and $H$ are never formed. Matrix vector products $\hat{H}v$ are computed as $u \leftarrow Av$, followed by $\hat{H}v \leftarrow (1+\beta)v - \alpha u$; only an oracle returning matrix vector products with $A$ is needed, i.e., the process is 'matrix free'.*

### 2.1 The Power Method with Momentum

The Power Method (PM) can be used to approximate the dominant eigenvalue and corresponding eigenvector of a matrix $M \in \mathbb{R}^{n \times n}$: $(\nu_1, u_1)$. A variant of the PM is developed in Xu et al. (2018), that can lead to faster approximation of the eigen-pair. The method, characterised by update (A) in Xu et al. (2018) (henceforth referred to as the PM with Momentum (PMM)), includes a momentum term with fixed momentum parameter $\beta$. Given an initial point $w^{(0)} \in \mathbb{R}^n$, and setting $w^{(-1)} \equiv 0$, for $k \geq 0$ the iterate update is

$$w^{(k+1)} = Mw^{(k)} - \beta w^{(k-1)}. \tag{10}$$

Setting $\beta = 0$ in equation 10 recovers the PM. Typically, the dominant eigenvalue is approximated using the Rayleigh quotient

$$\nu_1^{(k)} = ((w^{(k)})^T M w^{(k)})/\|w^{(k)}\|_2^2, \tag{11}$$

and a normalization step is often included in the algorithm to ensure that the approximate eigenvector does not grow too large (thereby avoiding numerical instabilities).

Convergence results for the PM can be found in many good textbooks (see, for example, Theorem 27.1 in Trefethen & David Bau (1997), and results in Demmel (1997); Golub & Loan (1996)). It is important to note that convergence of the PM requires the initial vector $w^{(0)}$ to contain a component in the direction of the dominant eigenvector, i.e., $u_1^T w^{(0)} \neq 0$. Convergence of the PMM was established in Xu et al. (2018), and it was shown that the optimal choice for the momentum parameter $\beta$, for matrix $M$, is $\beta = (\nu_2(M))^2/4$, where $\nu_2(M)$ denotes the second largest eigenvalue of $M$ (in magnitude). Thus, the PMM can be difficult to tune because a good approximation to $\nu_2(M)$ is often unknown a priori.

For GDM the gradients evolve as in (7), but this is equivalent to the evolution of the eigenvector approximation (10) for the PMM with $M = \hat{H}$ and $w^{(0)} = g^{(0)}$. So GDM with a fixed step size is implicitly running the PMM on the gradients and hence, the connection between GDM and the PMM is confirmed.

Because the Power Method with Momentum is an algorithm for approximating the dominant eigenvalue and corresponding eigenvector of a matrix, the observation that the gradients are computed via a PMM

iteration shows that the gradients in GDM are aligning in the direction of the dominant eigenvector of $\hat{H}$. Subsequently, one has access to approximations to the dominant eigen-pair of $\hat{H}$. The following lemma describes the eigen-pairs of $\hat{H}$, and how they are related to the eigen-pairs of $A$.

**Lemma 2** (Eigenvalues of $\hat{H}$). *Let Assumption 1 hold, and fix $\alpha \in (0, \frac{1}{L}]$ and $\beta \in [0, 1]$. Then, the eigen-pairs of $\hat{H}$ in (6), are $(\nu_i, v_{n-i+1})$, where*

$$\nu_i = 1 + \beta - \alpha\lambda_{n-i+1}, \quad for\ all\ \ i = 1, \ldots, n. \tag{12}$$

*Proof.* By Lemma 1, the eigen-pairs of $A$ are $(\lambda_i, v_i)$, for $i = 1, \ldots, n$, where $\lambda_n \leq \cdots \leq \lambda_1$. Thus, the eigenvalues of $\hat{H} = (1 + \beta)I - \alpha A$ are $\nu_1 = 1 + \beta - \alpha\lambda_n \geq \cdots \geq \nu_n = 1 + \beta - \alpha\lambda_1$. Finally, note that the ordering of the eigenvectors is reversed, so the $i$th eigenvalue of $\hat{H}$, $\nu_i$, is associated with the $(n - i + 1)$th eigenvector of $A$, $v_{n-i+1}$. □

By Lemma 2, the dominant eigen-pair of $\hat{H}$ is $(\nu_1, v_n)$, where $\nu_1 = 1 + \beta - \alpha\lambda_n$, while the leftmost eigen-pair of $A$ is $(\lambda_n, v_n)$, where $\lambda_n = \frac{1}{\alpha}(1 + \beta - \nu_1)$. That is, assuming that $v_n^T g^{(0)} \neq 0$, Lemmas 1 and 2 show that the gradients in GD with a fixed step size are converging in the direction of the leftmost eigenvector of $A$, and an approximation to $\lambda_n$ is also readily obtained. Therefore, valuable curvature information related to $A$ is implicitly available via Gradient Descent with Momentum, with a fixed step size. Note also that if $\lambda_n \geq 0$, then the spectral radius of $\hat{H}$ in (6) is $\rho(\hat{H}) = 1 + \beta - \alpha\lambda_n \leq 1 + \beta$.

To the best of our knowledge, this is the first work to formally connect GDM and the PMM. However, note that mathematical expressions similar to equation 9 can be found in many textbooks on linear algebra and optimization. For example, (3.4) in van der Vorst (2003) gives an expression for the relationship between the residuals (i.e., negative gradients) for Richardson's method, which is similar to equation 9. Similarly, see the error reduction formula (van der Vorst, 2003, p.24). See also the development in Chapter 2 of Greenbaum (1997), starting from expression (2.1) used with $M = \lambda_1 I$. Despite similar relationships appearing in several places, we have not seen it explicitly written that GDM with fixed step size and momentum parameter is equivalent to the PMM acting on the gradients, nor making the connection with the eigenvalues of the original matrix $A$. Possibly this is because expressions such as equation 9 are often made as a initial argument, providing motivation for more sophisticated methods, which are shown to have better convergence properties than GD. (For example, expressing the residual at iteration $k + 1$ in terms of a polynomial multiplied by the residual at iteration $k$, and showing that minimizing the polynomial with respect to different norms generates the iterates of various algorithms.)

## 2.2 GDM with Eigen-pair Approximation

For completeness, Algorithm 1 is an example of how GDM and the PMM can be combined to give both the solution to (1) *and* an explicit approximation to the leftmost eigen-pair of $A$.

---

**Algorithm 1** GDM (Eigen-revealing implementation)

---

1: **Input:** $x^{(0)} \in \mathbb{R}^n$, $x^{(-1)} = x^{(0)}$, $\alpha \in (0, \frac{1}{L}]$, $\beta \in [0, 1]$
2: **Initialize:** $g^{(0)} = Ax^{(0)} - b = g^{(-1)}$
3: **for** $k = 0, 1, 2, \ldots$ **do**
4:     $x^{(k+1)} = x^{(k)} - \alpha g^{(k)} + \beta(x^{(k)} - x^{(k-1)})$
5:     $w^{(k)} = Ag^{(k)}$
6:     $u^{(k)} = (1 + \beta)g^{(k)} - \alpha w^{(k)}$
7:     $\nu_1^{(k)} = (g^{(k)})^T u^{(k)} / \|g^{(k)}\|_2^2$
8:     $g^{(k+1)} = u^{(k)} - \beta g^{(k-1)}$
9:     $\lambda_n^{(k)} = (1 + \beta - \nu_1^{(k)})/\alpha$
10: **end for**

---

Step 4 in Algorithm 1 is the standard GDM update (equivalent to (4)), while Steps 5, 6 and 8 update the gradient. Steps 7 and 9 compute the approximate eigen-information, with the Rayleigh quotient used in

Step 7 because it "is a quadratically accurate estimate of an eigenvalue" (Trefethen & David Bau, 1997, p.204). The main computational cost is a single matrix vector product in Step 5. The eigen-pair approximation $(\lambda_n, v_n) \approx ((1 + \beta - \nu_1^{(k)})/\alpha, g^{(k)})$ is 'free', costing several inner products only. If $\nu_1^{(k)} = 1$ for any $k \geq 0$, then $\lambda_n = 0$, and one concludes that $A$ is singular. If $\nu_1^{(k)} > 1$ for any $k \geq 0$, then $\lambda_n < 0$, and therefore $A$ is indefinite.

### 2.2.1 Theory

Algorithm 1 is simply GDM coupled with an eigenpair $(\lambda_n, v_n)$ approximation. Thus, all theoretical results for GDM (convergence rate, complexity, etc) follows directly from the existing literature.

Note that the PMM typically includes a normalisation step to ensure that the eigenvector approximation does not grow too large, which could lead to numerical issues. However, for gradient descent with an appropriate $\alpha$ and $\beta$, the norm of the gradient decreases as the iterates move toward the optimal solution, and scaling the gradient is not desirable. (The norm of the gradient is often used as a stopping condition.)

**Remark 2.** *Convergence of the PM requires the initial vector to contain a component in the direction of the dominant eigenvector. Thus, for $v_n$ to be recovered, Algorithm 1 must be initialized so that $v_n^T g^{(0)} \neq 0$, i.e., $g^{(0)}$ must contain a component in the direction of the leftmost eigenvector, $v_n$.*

For a symmetric PD matrix, applying the PM ($\beta = 0$) to the shifted matrix $A - LI$ allows recovery of the smallest eigenvalue of $A$ $\lambda_n (\equiv \mu)$, i.e., the dominant eigenvalue of $A - LI$ is $|\mu - L|$. Clearly, the rate of convergence of the Power Method applied to the shifted matrix is $\left|\frac{\lambda_{n-1} - L}{\mu - L}\right|^{2k}$. For PD $A$, this matches the rate of GD-EIG because for $H$ in equation 5, the rate of convergence of the PM is

$$\left|\frac{\nu_2}{\nu_1}\right|^{2k} = \left|\frac{1 - \frac{\lambda_{n-1}}{L}}{1 - \frac{\mu}{L}}\right|^{2k} = \left|\frac{L - \lambda_{n-1}}{L - \mu}\right|^{2k}. \tag{13}$$

The ratio in equation 13 is smaller (i.e., convergence is faster) when the gap between $\lambda_{n-1}$ and $\lambda_n \equiv \mu$ is larger. For a computable error bound on the accuracy of the eigenvalue approximation given by the Power Method, see Theorem 8.1.13 and the comment on p.408 in Golub & Loan (1996).

Algorithm 1 is an eigen-pair revealing implementation of GDM; given a fixed step size $\alpha \in (0, \frac{1}{L}]$, GDM returns an approximate solution to equation 1 *and* an approximation to the leftmost eigenvalue $\lambda_n$. There are many algorithms that may benefit from this information. For example, a trust region subproblem sometimes requires eigen-information, so Algorithm 1 could be used as the sub solver (see, for example, discussion of the 'hard case' in (Erway et al., 2020, Section 2.3), (Nocedal & Wright, 2006, p.87)); accelerated gradient methods typically use the strong convexity constant, so Algorithm 1 could be used as a warm start to approximate $\mu \equiv \lambda_n$ before switching to an accelerated gradient method Nesterov (1983); and it may be helpful in applications involving the solution a sequence of equations with a slowly changing right hand side, or as a means of running a deflation technique Bellavia et al. (2013).

Further to these, an estimate of the left-most eigen-pair $(\lambda_n, v_n)$ could be used as part of a second order convergence test to determine whether a stationary point is a local minimizer; coupled with an approximation to $L \equiv \lambda_1$, $\mu \equiv \lambda_n$ provides an estimate of the condition number of $A$, and in turn, iteration complexity results exist for GD which involve an estimate of the condition number, so GD-EIG could be used to compute an estimate of the number of iterations required for convergence on-the-fly.

## 3  Behaviour of GD near saddle points via examples

Several works including Du et al. (2017) and Paternain et al. (2019), present examples showing that GD can take exponential time to escape saddle points, and they argue that alternatives to vanilla GD with a fixed step size may be preferable for nonconvex optimization problems. Those examples are revisited now, to show that incorporating eigen-information may be helpful.

### 3.1 Example of Paternain et al. (2019)

Paternain et al. (2019) presents the following illustrative example to show that GD can escape from saddle points, but does so slowly. Consider the function:

$$f_\sigma(x) = \tfrac{1}{2}x_1^2 - \tfrac{\sigma}{2}x_2^2 = \tfrac{1}{2}x^T A x, \tag{14}$$

where $A = \text{diag}(1, -\sigma)$, so the eigenvalues are $\lambda_1 = 1$ and $\lambda_2 = -\sigma$ (it is assumed that $\lambda_1 > 0 > \lambda_2$, i.e., $\sigma > 0$). Now suppose that GD is applied to minimize the function in equation 14 using a fixed step size $\alpha = 1/\lambda_1 = 1$ (no momentum, $\beta = 0$). Given an initial point $x^{(0)} = (x_1^{(0)}, x_2^{(0)})^T$ and corresponding gradient $g^{(0)} = Ax^{(0)} = (x_1^{(0)}, -\sigma x_2^{(0)})^T$, for all $k \geq 1$, the iterates and gradient evolve as

$$x_1^{(k)} = 0, \qquad x_2^{(k)} = (1+\sigma)^k x_2^{(0)}, \tag{15}$$

and

$$g_1^{(k)} = 0, \qquad g_2^{(k)} = -\sigma(1+\sigma)^k x_2^{(0)}. \tag{16}$$

As $\sigma \to 0$, (i.e., as the problem becomes increasingly ill-conditioned), it takes longer to escape the saddle point. As noted in Paternain et al. (2019), this example demonstrates that GD with a fixed step size will take exponential time to escape from the saddle point with the rate $(1+\sigma)$.

On the other hand, the iterates of their novel Non-Convex Newton (NCN) method (Paternain et al. (2019)) evolve as

$$x_1^{(k)} = 0, \qquad x_2^{(k)} = 2^k x_2^{(0)},$$

so NCN also takes exponential time to escape the saddle point, but at the better rate of 2, which is *independent of $\sigma$*, (i.e., the rate is independent of problem conditioning).

Now, recalling $A$ in equation 14, the dominant eigenvalue of $H \overset{5}{=} I - \tfrac{1}{1}A$, for $k \geq 1$ can be approximated using the Rayleigh quotient as

$$\nu_1^{(k)} = \frac{(g^{(k)})^T H g^{(k)}}{(g^{(k)})^T g^{(k)}} = \frac{(g^{(k)})^T g^{(k+1)}}{(g^{(k)})^T g^{(k)}} = \frac{\sigma^2(1+\sigma)^{2k+1}(x_2^{(0)})^2}{\sigma^2(1+\sigma)^{2k}(x_2^{(0)})^2} = 1 + \sigma. \tag{17}$$

But the dominant eigenvalue of $H$ *is* $1 + \sigma$, so the approximation in equation 17 is, in fact, exact. Importantly, $\nu_1^{(1)} > 1$, so it is recognised that the leftmost eigenvalue of $A$ is negative, and it can be explicitly recovered as $\lambda_2^{(1)} = \lambda_1(1 - \nu_1^{(1)}) = -\sigma$. By (16) and (17), the error $\delta = \|\nu_1^{(1)} g^{(1)} - g^{(2)}\|^2 = 0$ confirms that the approximation is exact, as is the corresponding eigenvector, $g^{(2)} \equiv v_2$.

In summary, after exactly 2 iterations of GDM (implemented as Algorithm 1) with an fixed step length $\alpha = 1/\lambda_1$, the leftmost eigen-pair of $A$ is recovered, and it is known that the original problem is nonconvex because $\lambda_2 < 0$ (so there is a saddle point). With vanilla gradient descent, this valuable curvature information is wasted because it is never explicitly computed. *Can this eigen information be used to enrich gradient descent?* This question will be investigated further in Section **??**, but let us try to gain some intuition to address this question now using this example in $\mathbb{R}^2$. Taking an initial step of size $1/\lambda_1$ reduced the problem to a 1-dimensional subspace ($g_1^{(k)} = 0$ in equation 16). The iterate and gradient dynamics in equation 15 and equation 16 show that keeping the step $\alpha = 1$ is then 'too short' for iterations $k \geq 2$ of this problem, and results in very slow progress. Instead, can equation 17 be used to guide the step size choice in iteration $k = 2$? Let $\alpha^{(2)}$ denote the potential step length. Then

$$x_2^{(3)} = x_2^{(2)} - \tfrac{1}{\alpha^{(2)}}g_2^{(2)} = (1+\sigma)^2 x_2^{(2)} - \tfrac{1}{\alpha^{(2)}}(-\sigma(1+\sigma)^2 x_2^{(2)}) = \left(1 + \tfrac{\sigma}{\alpha^{(2)}}\right)(1+\sigma)^2 x_2^{(0)}.$$

Setting $\alpha^{(2)} = \lambda_2 = -\sigma$ leads directly to the saddle point $(0,0)$. On the other hand, choosing $\alpha^{(2)} = |\lambda_2| = \sigma$, gives $x_2^{(3)} = 2(1+\sigma)^2 x_2^{(2)}$. Because $\sigma > 0$, $2(1+\sigma)^2 > 2$, so that the 'escape rate' is better using this eigen-information. Thus, using a first-order method, enriched with curvature information, the step size can be adapted so that the iterates escape the saddle point faster than vanilla GD.

### 3.2 Example of Du et al. (2017)

Du et al. (2017) present the following example. Consider a two-dimensional function $f$ with a strict saddle point at (0,0). Suppose that inside the neighbourhood $U = [-1, 1]^2$ of the saddle point, the function is locally quadratic $f(x_1, x_2) = x_1^2 - x_2^2$. (Note that this can be written equivalently as $f(x_1, x_2) = \frac{1}{2} x^T A x$ where $A = \mathrm{diag}(2, -2)$ so the eigenvalues are $\lambda_1 = 2$ and $\lambda_2 = -2$.) For GD with $\alpha = 1/4$ the coordinates are updated as

$$x_1^{(k+1)} = \tfrac{1}{2} x_1^{(k)} \quad \text{and} \quad x_2^{(k+1)} = \tfrac{3}{2} x_2^{(k)}.$$

Du et al. (2017) argues that if GD is initialized uniformly within the exponentially thin band $[-1, 1] \times [-(\frac{3}{2})^{-\exp(1/\epsilon)}, (\frac{3}{2})^{-\exp(1/\epsilon)}]$, (the width is $2(\frac{3}{2})^{-\exp(1/\epsilon)}$) then GD requires at least $\exp(1/\epsilon)$ iterations to leave the neighbourhood $U$ and escape the saddle point. Note that the suggested initialization band for the component $x_2^{(0)}$ is so thin that $x_2^{(0)} \approx 0$ even for relatively large values of $\epsilon$ (if $\epsilon = 0.2$ then $x_2^{(0)} = 10^{-27}$ and if $\epsilon = 0.1$ then $x_2^{(0)} = 0$ is returned by MATLAB).

Now consider the following. First, note that the step length is half what it could be to ensure convergence. That is, $\alpha = \frac{1}{4} = \frac{1}{2\lambda_1} = \frac{1}{c\lambda_1}$ with $c = 2$, whereas the arguments in Section ?? suggested to use the step size $\alpha = \frac{1}{\lambda_1} = \frac{1}{2}$. It can be seen (for this special case where the eigenvectors align in the coordinate directions) that the step is too short, because $x_1^{(1)} \neq 0$, $g_1^{(1)} \neq 0$, and because

$$g_1^{(k+1)} = 2 \cdot \left(\tfrac{1}{2}\right)^k x_1^{(0)} \quad \text{and} \quad g_2^{(k+1)} = 2 \cdot \left(\tfrac{3}{2}\right)^k x_2^{(0)}.$$

The eigenvalue approximation is

$$\nu_1^{(k)} = \frac{(g^{(k)})^T A g^{(k)}}{(g^{(k)})^T g^{(k)}} = \frac{(g^{(k)})^T g^{(k+1)}}{(g^{(k)})^T g^{(k)}} = \frac{(1/2)^{2k+1} (x_1^{(0)})^2 + (3/2)^{2k+1} (x_2^{(0)})^2}{(1/2)^{2k} (x_1^{(0)})^2 + (3/2)^{2k} (x_2^{(0)})^2}.$$

#### 3.2.1 Numerical Experiments

Several numerical experiments are presented on the problem described above to further investigate the behaviour of GD as $\epsilon$ and $\alpha$ are varied. All are initialized at $x^{(0)} = (1, (\frac{3}{2})^{-\exp(1/\epsilon)})$ for either $\epsilon = 0.1$ (i.e., $x_2^{(0)} = 0$) or $\epsilon = 0.5$ (i.e., $x_2^{(0)} \neq 0$). Recall that $\lambda_2^{(k)} = (1 - \nu_1^{(k)})/\alpha$ and define the error in the eigenvalue approximation to be $\delta^{(k)} = \|A g^{(k)} - \lambda_2^{(k)} g^{(k)}\|_2$.

**Numerical experiment 1.** Let $\epsilon = 0.1$ (so $x^{(0)} = (1, 0)^T$) and let $\alpha = 1/4$. Then for $k = 1$, Algorithm 1 gives $x^{(1)} = (0.5, 0)^T$, $\nu_1^{(1)} = 0.5$, $\lambda_1^{(1)} = -2$ and $\delta^{(1)} = 0$. Because $\delta^{(1)} = 0$ (the error is zero), an exact eigenvalue is found and $g^{(1)}$ points in the direction of the corresponding eigenvector. Then taking the step $x^{(2)} = x^{(1)} - \frac{1}{\lambda_2} g^{(1)} = (0, 0)^T$, i.e., the saddle point has been reached. The saddle point *cannot be escaped*, because there is no 'escaping direction'. Recall Remark 2, which explains that this is because $v_2^T g^{(0)} = 0$.

A stationary point has been found ($g^{(2)} = (0, 0)^T$), and to learn the nature of the stationary point, consider the following. Given the known eigen-pair, choose a new random point, $\tilde{x}^{(0)}$ say, (possibly close to the solution/origin) that is *not co-linear* with the known eigenvector. Then running 2 iterations of GD-EIG from $\tilde{x}^{(0)}$ with the step size $\alpha = |1/\lambda_2| = 1/2$ gives the complete information (see also numerical experiment 2).

**Numerical experiment 2.** Let $\epsilon = 0.5$ (so $x^{(0)} = (1, 0.05)^T$) and let $\alpha = 1/\lambda_1 = 1/2$. The iterates of Algorithm 1 are $x^{(1)} = (0, 0.1)^T$ with $\delta^{(1)} \neq 0$, and $x^{(2)} = (0, 0.1999)^T$ with $\nu_1^{(2)} = 2 \ (> 1)$, $\lambda_2^{(2)} = -2$, and $\delta^{(1)} = 0$, i.e., after 2 iterations of Algorithm 1 with $\alpha = 1/2$, both eigenvalues are known, and the eigenvector corresponding to the negative eigenvalue (i.e., an escaping direction) is known. Taking the step $\alpha^{(2)} = 1/\lambda_2 = -1/2$ leads directly to the saddle point.

**Numerical experiment 3.** Let $\epsilon = 0.5$ (so $x^{(0)} = (1, 0.05)^T$) and let $\alpha = 1/4$. The iterates of Algorithm 1 are reported in Table 1.

Table 1: Behaviour of GD-EIG on the above example with $\epsilon = 0.5$ and $\alpha = 1/4$.

| $k$ | 0 | 1 | 2 | 3 | 4 |
|---|---|---|---|---|---|
| $x_1^{(k)}$ | 1 | 0.5 | 0.25 | 0.125 | 0.0625 |
| $x_2^{(k)}$ | 0.05 | 0.075 | 0.1125 | 0.1687 | 0.2531 |
| $\nu_1^{(k)}$ | — | 0.5025 | 0.5220 | 0.6683 | 1.1456 |
| $\lambda_2^{(k)}$ | — | 1.9900 | 1.9120 | 1.3267 | -0.5823 |
| $\delta^{(k)}$ | — | 0.5984 | 0.8811 | 1.1350 | 0.7868 |

After $k = 4$ iterations the approximation is $\nu_1^{(4)} > 1$, which flags that $A$ has a negative eigenvalue, and consequently, $g^{(4)} = (0.1250, -0.5061)^T$ is a direction of negative curvature. Note that the error is nonzero $\delta^{(4)} \neq 0$, so even though the leftmost eigenvalue has not yet been located ($\lambda_2^{(k)} \neq \lambda_2$, $g^{(4)} \neq v_2$), Algorithm 1 has promptly identified that the matrix is indefinite. (After $k = 6$ iterations, $\lambda_2^{(6)} = -1.9731$ and $\delta^{(6)} = 0.1279$, and after $k = 10$ iterations, $\lambda_2^{(10)} = -2.0000$ and $\delta^{(10)} = 0.0078$.)

Given this information, rather than continuing with GD with a fixed step size, alternative strategies may include: (1) taking a step of size $\alpha^{(4)} = -1/\lambda_2^{(4)}$ (which would give $x^{(5)} = x^{(4)} - \alpha^{(4)} g^{(4)} = (-0.1522, 1.1221)^T$); (2) perform a line search along the direction of negative curvature; or (3) continue Algorithm 1 until $\delta^{(k)} \approx 0$.

## 4 Gradient Descent with a Kick (GD-Kick)

In this section a new eigen-enriched gradient descent algorithm is presented. As motivation, first let us study the dynamics of the gradient for GD. Throughout this section let Assumption 1 hold, and for simplicity suppose that the eigenvalues are distinct and nonzero (i.e., $A$ is nonsingular), and that $\beta = 0$ (no momentum).

### 4.1 Dynamics of GD

Denote the eigen-decomposition of $A \in \mathbb{R}^{n \times n}$ by $A = V \Lambda V^T$, where $V = \begin{bmatrix} v_1 & \cdots & v_n \end{bmatrix}$ and $\Lambda = \text{diag}(\lambda_1, \ldots, \lambda_n)$. Because $A$ is symmetric (Assumption 1), $V$ is orthogonal, and the eigenvectors form an orthonormal basis for $\mathbb{R}^n$. The eigen-decomposition is $H$ in equation 5 is

$$H = I - \alpha A = I - \alpha V \Lambda V^T = V(I - \alpha \Lambda)V^T = V \Lambda_H V^T, \tag{18}$$

where $\Lambda_H = \text{diag}(1 - \alpha \lambda_1, \ldots, 1 - \alpha \lambda_n)$ and $A$ and $H$ share the common eigenvectors given in $V$. Then, for any iteration $k \geq 0$ of GD with fixed $\alpha$, the gradient is

$$g^{(k)} \overset{(9)}{=} H^k g^{(0)} \overset{(18)}{=} V \Lambda_H^k V^T g^{(0)} = \sum_{i=1}^{n} (1 - \alpha \lambda_i)^k (v_i^T g^{(0)}) v_i. \tag{19}$$

The dynamics of the gradient in equation 19 show the following. Suppose that $v_i^T g^{(0)} \neq 0 \; \forall i$, and suppose that $\lambda_n > 0$ (i.e., let $A$ be PD). Then, for $g^{(k)} \to 0$ as $k \to \infty$, all the coefficients $(1 - \alpha \lambda_i)^k$ must tend to zero. If $\alpha = 1/L \; (\equiv 1/\lambda_1)$, then $(1 - \alpha \lambda_1)^1 = 0$, and $(1 - \alpha \lambda_i) = (1 - \frac{\lambda_i}{L}) < 1$ for $i = 2, \ldots, n$, so $(1 - \alpha \lambda_i)^k \to 0$. So, after 1 iteration of GD with the step length $\alpha = 1/L$, the component of the gradient in the direction of $v_1$ is eradicated, and the gradient lies in the $n - 1$-dimensional subspace spanned by the remaining eigenvectors $\{v_2, \ldots, v_n\}$. The term $(1 - \alpha \lambda_n)^k$ tends to zero the most slowly, confirming from a different viewpoint that the gradient aligns in the direction of the leftmost eigenvector as $k \to \infty$.

**Remark 3.** *As is known, if the eigenvalues $\lambda_1, \ldots, \lambda_n$ are known, taking precisely n steps of gradient descent, but with steplength $1/\lambda_k$ for iterations $k = 1, \ldots, n$, gives a zero gradient.[1], i.e., equation 19 shows that once an eigen-component has been cancelled out from the gradient, it will never re-enter the gradient (in exact arithmetic).*

---

[1] In fact, in exact arithmetic, the order of the step-lengths does not matter, only that exactly one step of size $1/\lambda_k$ for each $k = 1, \ldots, n$ is taken. However, for numerical reasons, the order $1/\lambda_1, \ldots, 1/\lambda_n$ is sensible.

If $A$ is indefinite, then for all $\lambda_j < 0$, $(1 - \frac{\lambda_j}{L}) > 1$, so continuing with the fixed step size $\alpha = 1/L$ causes the coefficients in equation 19 corresponding to negative eigenvalues to grow, and the iterates will escape the saddle point eventually.

### 4.2 A locally adaptive eigen-enriched GD type method

An eigen-enriched GD algorithm called Gradient Descent with a Kick (GD-Kick) is presented now. GD-Kick is motivated by the previous discussion, which showed that a fixed step size is typically associated with the slow progress of vanilla GD, and the question of whether the 'free', implicitly gathered, approximate and locally adaptive curvature information can be used to improve the practical behaviour of GD. GD-Kick uses a 'base' algorithm of GD with a fixed step size, but every $s$ iterations, the algorithm attempts to take a locally adaptive longer step (a 'kick'), where the step size is an approximation to the reciprocal of the leftmost eigenvalue. GD-Kick is presented as Algorithm 2.

---

**Algorithm 2** GD-Kick

---

1: **Input:** Initial point $x^{(0)}$, $\alpha \in (0, 1/L]$, parameter $s \geq 1$.
2: **for** $k = 0, 1, 2, \ldots$ **do**
3:     $x^+ = x^{(k)} - \alpha g^{(k)}$
4:     **if** $\mod (k, s) \equiv 0$ **then**
5:         $\nu_1^{(k)} = ((g^{(k)})^T A g^{(k)})/\|g^{(k)}\|_2^2$
6:         $\lambda_n^{(k)} = (1 - \nu_1^{(k)})/\alpha$
7:         **if** $\lambda_n^{(k)} \neq 0$ **then**
8:             $x^{++} = x^{(k)} - |1/\lambda_n^{(k)}| g^{(k)}$
9:             **if** $f(x^{++}) < f(x^+)$ **then**
10:                $x^+ \leftarrow x^{++}$
11:             **end**
12:         **end**
13:     **end**
14:     $x^{(k+1)} = x^+$
15: **end for**

---

Section 4.1 showed that running GD with $\alpha = 1/L$ eliminates the component of the gradient in the direction of the dominant eigenvector, and if $A$ is PD, the remaining components in equation 19 shrink, although components with $\lambda_1 \gg \lambda_i$ go to zero very slowly. If $\mod (k, s) \equiv 0$, then Steps 5 and 6 use the Rayleigh quotient to approximate $\lambda_n^{(k)}$, and a long step (because $1/\lambda_n^{(k)} \geq 1/\lambda_1$) is attempted in Step 8. Considering equation 19 if $\lambda_n^{(k)} \approx \lambda_i$ for any $i$, then the corresponding component of the gradient becomes closer to zero than if the fixed step were taken. Thus, taking this larger step will hopefully be beneficial for shrinking the components of the gradient corresponding to smaller eigenvalues. Equally, this larger step can cause components of the gradient corresponding to larger eigenvalues to increase, so the step is only accepted (see Step 9) if the function value reduces more than for the fixed step $\alpha$. (Note that the norm of the gradient is not prohibited from increasing.) The algorithm then returns to the fixed step size $\alpha$, and this 'fixed step then kick' process repeats.

**Remark 4.** *Algorithm 2 does not restrict $\alpha = 1/L$. Indeed any $\alpha \in (0, 1/L]$ is allowed, but if $\alpha \neq 1/L$ then the component of $g$ in the direction of the dominant eigenvector is not eliminated in the first iteration (it simply shrinks).*

If $A$ is indefinite, then $\lambda_n^{(k)}$ still approximates the leftmost eigenvalue, but now the approximation could be negative, so Step 8 uses the absolute value of the reciprocal of $\lambda_n^{(k)}$ as the step size. As discussed previously, this will hopefully lead to faster evasion of the saddle point. Note that, if the goal was to locate a stationary point (for example, if one wished to solve $Ax = b$, rather than equation 1), then dropping the absolute values and taking the step size $1/\lambda_n^{(k)}$ gives a kick toward the saddle.

Two safeguards are included in GD-Kick. Firstly, if $A$ is singular, then it is possible that $\lambda_n^{(k)} = 0$, in which case the step in Step 8 is undefined. Hence, Step 7 is included to ensure that a 'kick' step is only considered if $\lambda_n^{(k)} \neq 0$. Secondly, a longer 'kick' step is only accepted if it leads to a greater reduction in the function value than a corresponding fixed step size, thus ensuring algorithm convergence (see Theorem 1).

GD-Kick makes use of the 'freely available' eigen-information, so it is no more expensive than vanilla GD; the dominant cost is one matrix vector product per iteration. Moreover, the connection with the PM still holds, because GD-Kick can be viewed as vanilla GD with restarts. Thus, GD-Kick still provides approximations to $\lambda_n$, and the gradient still aligns in the direction $v_n$ as $k \to \infty$.

The following result gives convergence properties for GD-Kick (Algorithm 2). Note that here, $f$ is not necessarily quadratic.

**Theorem 1.** *Let $f$ satisfy equation 3, let $\alpha \in (0, 1/L]$ and let $s \geq 1$. If GD-Kick (Algorithm 2) is applied to $\min_{x \in \mathbb{R}^n} f(x)$, then for all $k \geq 0$*

    *1. if $f$ is strongly convex then*

$$f(x^{(k)}) - f^* \leq (1 - \alpha\mu)^k (f(x^{(0)}) - f^*); \tag{20}$$

    *2. or if $f$ is convex and bounded below*

$$f(x^{(k)}) - f^* \leq \frac{\|x^{(0)} - x^*\|_2^2}{2\alpha k}; \tag{21}$$

    *3. or if $f$ is nonconvex and bounded below by $f^*$ then $\|g^{(k)}\|_2 \to 0$ as $k \to \infty$.*

*Proof.* For any $k \geq 0$ with $\mod(k,s) \not\equiv 0$, Steps 3 and 14 show that GD-Kick updates the iterate using a constant step size $\alpha \in (0, \frac{1}{L}]$. The only other possibility occurs if both $\mod(k,s) \equiv 0$ *and* $\lambda_n^{(k)} \neq 0$ hold, in which case a long step $x^{++}$ is attempted in Step 8. The long step is accepted (Step 10) only if it results in a lower function value than for the constant step $\alpha$. Thus, for any $k \geq 0$ it must hold that

$$
\begin{aligned}
f(x^{(k+1)}) &\leq \min\{f(x^{(k)} - \alpha\nabla f(x^{(k)})), f(x^{(k)} - |1/\lambda_n^{(k)}|\nabla f(x^{(k)}))\} \\
&\leq f(x^{(k)} - \alpha\nabla f(x^{(k)})) \\
&\leq f(x^{(k)}) + (\nabla f(x^{(k)})^T(x^{(k+1)} - x^{(k)}) + \frac{L}{2}\|x^{(k+1)} - x^{(k)}\|_2^2 \\
&\leq f(x^{(k)}) - \alpha(1 - \frac{\alpha L}{2})\|\nabla f(x^{(k)})\|_2^2 \\
&\leq f(x^{(k)}) - \frac{\alpha}{2}\|\nabla f(x^{(k)})\|_2^2, \tag{22}
\end{aligned}
$$

where the last step holds because $1 - \frac{\alpha L}{2} \geq 1/2$ whenever $\alpha \in (0, 1/L]$. Now, equation 22 can be combined with strong convexity and a recursion argument to give equation 20.

To show equation 21, combining equation 22 with convexity $f(x^{(k)}) \leq f^* + (\nabla f(x))^T(x^{(k)} - x^*)$, and completing the square gives $f(x^{(k+1)}) \leq f^* + \frac{1}{2\alpha}(\|x^{(k)} - x^*\|_2^2 - \|x^{(k+1)} - x^*\|_2^2)$. Summing over the iterations gives

$$\sum_{i=1}^{k}(f(x^{(i)}) - f^*) \leq \frac{1}{2\alpha}(\|x^{(0)} - x^*\|_2^2 - \|x^{(k)} - x^*\|_2^2) \leq \frac{1}{2\alpha}\|x^{(0)} - x^*\|_2^2.$$

Thus, $f(x^{(k)})$ is nonincreasing, so dividing the above through by $k$ and noticing that $f(x^{(k)}) - f^* \leq \frac{1}{k}\sum_{i=1}^{k}(f(x^{(i)}) - f^*)$ gives equation 21.

Finally, following equation 22, the arguments in (Nesterov, 2004, p.27 and (1.2.14)) show 3. $\qquad\square$

Note that the proof of Theorem 1 does not rely upon $f$ being quadratic; $f$ is only assumed to be (strongly) convex, twice continuously differentiable, and have a Lipschitz continuous gradient. These are standard assumptions in the literature. Thus, Algorithm 2 is guaranteed to converge when applied to a general function

with the aforementioned properties. Simply adjust Algorithm 2 by replacing $g^{(k)} \leftarrow \nabla f(x^{(k)})$ (i.e., replace $g^{(k)}$ by the derivative of the function $f$) and $A \leftarrow G^{(k)}(\approx \nabla^2 f(x^{(k)}))$ (i.e., replace $A$ by an approximation to the Hessian of the function $f$). In that case Step 5 still computes an approximation to an eigenvalue of $G^{(k)}$ but it is no longer guaranteed to be an approximation the left most eigenvalue.

**Remark 5.** *If $f$ is the quadratic function in equation 1, then Theorem 1 applies as follows. If $A$ is symmetric and positive definite, then Theorem 1(1) holds; if $A$ is indefinite, then equation 1 is unbounded below, so GD and GDKick diverge (i.e., Theorem 1 does not apply), and if $A$ is symmetric and positive semi-definite, then if $b \in \mathrm{col}(A)$ Theorem 1(2) holds, while if $b \notin \mathrm{col}(A)$ then equation 1 is unbounded below and so GD and GDKick diverge.*

### 4.3 Choice of step length

Section 4.1 showed that if $\alpha = 1/L$, then $(1 - \alpha\lambda_1) = 0$ (the first coefficient in the sum in equation 19 is zero after 1 iteration), so that for $k \geq 2$, a more aggressive step length could be taken (while preserving convergence). Thus, consider the following 'smart' initialization strategy for GD with a fixed step size. Choose an initial point $x^{(-1)} \in \mathbb{R}^n$ and then set $x^{(0)} = x^{(-1)} - \frac{1}{\lambda_1}(Ax^{(-1)} - b)$, which ensures that $v_1^T g^{(0)} = 0$, i.e., the first component in the sum equation 19 corresponding to the direction $v_1$ is eliminated. Now, suppose that $\alpha = 2/\lambda_1$. Then, for all $\lambda_j > 0$, $|1 - \frac{2\lambda_j}{\lambda_1}| < 1$ so that $|1 - \frac{2\lambda_j}{\lambda_1}|^k \rightarrow 0$ as $k \rightarrow \infty$. Therefore, GD with the initialization above, and the fixed step size $\alpha = 2/L$ is guaranteed to converge whenever $A$ is positive definite. This is summarized in the following theorem.

**Theorem 2.** *Let $A$ be symmetric positive definite, let Assumption 1 and let $r := \max_{2 \leq i \leq n}(1 - \frac{2\lambda_i}{L})^2 < 1$. If gradient descent with the fixed step size $\alpha = 2/L$ is applied to (1) using the initialization and $x^{(0)} = x^{(-1)} - \frac{1}{L}(Ax^{(-1)} - b)$ for some $x^{(-1)} \in \mathbb{R}^n$, then $\|g^{(k)}\|_2^2 \leq r^k \|g^{(0)}\|_2^2$.*

*Proof.* For simplicity (and w.l.o.g), assume that $A = \Lambda$ where $\Lambda$ is a diagonal matrix, and that $b = 0$ in equation 1. Now, let $x^{(-1)} \in \mathbb{R}^n$ and set $x^{(0)} = x^{(-1)} - \frac{1}{L}g^{(-1)}$. Then,

$$g^{(0)} = (I - \tfrac{1}{L}\Lambda)g^{(-1)} = \begin{bmatrix} 0 & & & \\ & (1 - \frac{\lambda_2}{L}) & & \\ & & \ddots & \\ & & & (1 - \frac{\lambda_n}{L}) \end{bmatrix} g^{(-1)} = \begin{bmatrix} 0 \\ (1 - \frac{\lambda_2}{L})g_2^{(-1)} \\ \vdots \\ (1 - \frac{\lambda_n}{L})g_n^{(-1)} \end{bmatrix}, \tag{23}$$

i.e., the first component of $g^{(0)}$ is zero. Now, equation 19 simplifies to

$$\|g^{(k)}\|_2^2 = \sum_{i=1}^n (1 - \alpha\lambda_i)^{2k}(g_i^{(0)})^2 = \sum_{i=2}^n (1 - \alpha\lambda_i)^{2k}(g_i^{(0)})^2 = \sum_{i=2}^n (1 - \tfrac{2\lambda_i}{L})^{2k}(g_i^{(0)})^2. \tag{24}$$

Then if $r := \max_{2 \leq i \leq n}(1 - \frac{2\lambda_i}{L})^2 \; (< 1)$,

$$\|g^{(k)}\|_2^2 = \sum_{i=2}^n (1 - \tfrac{2\lambda_i}{L})^{2k}(g_i^{(0)})^2 \leq r^k \sum_{i=2}^n (g_i^{(0)})^2 \leq r^k \|g^{(0)}\|_2^2. \tag{25}$$

$\square$

If $A$ is indefinite, then for all $\lambda_j < 0$, $|1 - \frac{2\lambda_j}{\lambda_1}| > 1$ so these coefficients grow as $k \rightarrow \infty$, and because $|1 - \frac{2\lambda_j}{\lambda_1}| > |1 - \frac{\lambda_j}{\lambda_1}|$ the more aggressive step size $\alpha = 2/L$ leads to faster escape from the saddle point.

It is known that the step length $2/(L + \mu)$ can be used for GD when $A$ is positive definite. However, $\mu$ is often unknown when the algorithm is initialized, while the strategy above does not require knowledge of $\mu$. Also note that $2/L > 2/(L + \mu)$, i.e., it is a larger step.

**Remark 6.** *The fixed step $\alpha = 2/L$ can be used in GD-Kick (Algorithm 2). Simply take $\alpha = 1/L$ whenever $\mathrm{mod}\,(k, s) \equiv 1$, and then $\alpha = 2/L$ otherwise.*

## 5    Numerical Experiments

In this section, several experiments are presented to verify the results described in this work. In particular, experiments are presented showing that (1) an approximation to the leftmost eigen-pair can be recovered in practice; (2) that the fixed step length $\alpha = 2/L$, combined with the appropriate initialization, leads to favourable practical performance compared with the fixed step length $\alpha = 1/L$; and (3) that GD-Kick has improved practical behaviour compared with vanilla GD with a fixed step size.[2]

### 5.1    Investigating GD-Kick

The performance of GD-Kick is studied now. A symmetric positive definite matrix $A \in \mathbb{R}^{1000 \times 1000}$ was generated, as well as the optimal solution $x^* \in \mathbb{R}^{1000}$, with $b$ computed as $b = Ax^*$. For all experiments, the step size $\alpha^{(k)} = 2/L$ was used, coupled with the appropriate initialisation. The experiment compares GD with a fixed step size, with GD-Kick with 2 different choices of $s$, and the algorithms were also compared with the Accelerated Gradient Method as a benchmark. (Recall that the Accelerated Gradient Method requires knowledge of $\mu \equiv \lambda_n > 0$ in advance, which is not assumed for GD-Kick.) Figure 1 shows the results.

The top left plot shows the evolution of the function value error $f(x^{(k)}) - f^*$, the top middle plot shows the step size for GD-Kick, when $s = 20$ and $s = 100$, and the top right plot shows the evolution of the gradient norm $\|g^{(k)}\|_2^2$. Consider the top left plot. The blue line corresponds to 999 iterations of GD with the fixed step size $\alpha^{(k)} = 2/L$, coupled with the appropriate initial point. For the 1000th iteration, a 'kick' step (using the Rayleigh quotient) was used, which corresponds to the rapid decrease in the function value error in the last iteration. GD-Kick (the red and black lines) outperforms vanilla GD, and the choice $s = 20$ (19 iterations with a fixed step size, followed by a kick) is slightly better than $s = 100$, suggesting that it is advantageous to take a 'kick' step more often. Note that the accelerated gradient method (the green line) performs the best, but it requires prior knowledge of $\mu$.

Notice that when $\mod(k, s) \equiv 1$, the step is $\alpha = 1/L$, when $2 \leq \mod(k, s) \leq s - 1$ one takes $\alpha = 2/L$, and when $\mod(k, s) \equiv 0$, a long step $\alpha = 1/|\lambda_n^{(k)}|$, is attempted. The top middle plot shows this pattern, and also shows that the 'kick' helps the algorithm to make faster progress toward the solution.

The top right plot shows the evolution of the gradient norm. For GD with the fixed step size, the curve is smooth, and decreases slowly. For GD-Kick, with either choice of $s$, the norm of the gradient decreases more rapidly. Notice that at every $s$th iteration, the norm of the gradient increases, corresponding to the 'kick'. These kick steps also correspond to the larger decrease in the function value, giving a 'staircase' pattern.

The experiment described above was repeated for a matrix $A$ that is only positive semi-definite (50 eigenvalues were set to zero). Similar behaviour was observed for this problem. Note that the accelerated gradient method requires the objective function to be strongly convex, so it was not employed here.

### 5.2    Example from Du et al. (2017)

The work Du et al. (2017) presents an example of a smooth non-convex function $F : \mathbb{R}^2 \to \mathbb{R}$, which is made up quadratic regions that are joined using splines. This provided motivated to investigate the behaviour of the proposed algorithms on this non-quadratic problem. (Further details can be found in Du et al. (2017).)

Figure 2 shows the surface and contour plot of the function $F$ together with iterates of Gradient Descent (GD) run with the default step-size as suggested by authors in Du et al. (2017). To see why this problem is hard, one can investigate the size of gradient during the trajectory $\{x^{(k)}\}_{k=0}^\infty$.

In Figure 3 one observes that at the initial starting point, the norm of the gradient is already very small, and it takes almost 80 iterations to make a significant improvement in the function value $F(\cdot)$. Then, however, the iterates gets trapped close to a saddle point around $(4 \cdot e, 0)^T$, and the size of squared gradient becomes close to $10^{-28}$. To escape the saddle point, one suggested approach is to use Perturbed GD (PGD) Jin et al. (2017), which perturbs the current iterate in the case when the size of gradient is below some threshold for a

---

[2]The codes to reproduce numerical experiments are available at `https://github.com/Optimization-and-Machine-Learning-Lab/gradient_descent_and_power_method.git`

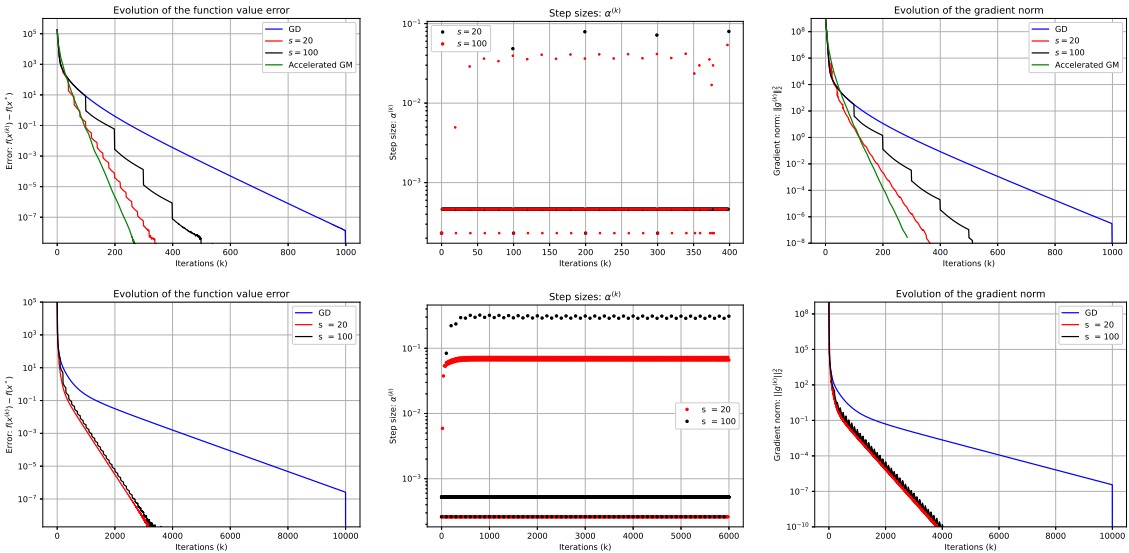

Figure 1: Gradient Descent with a kick (GD-Kick). The top row corresponds to a positive definite $A$, while the bottom row corresponds to a positive semi-definite $A$.

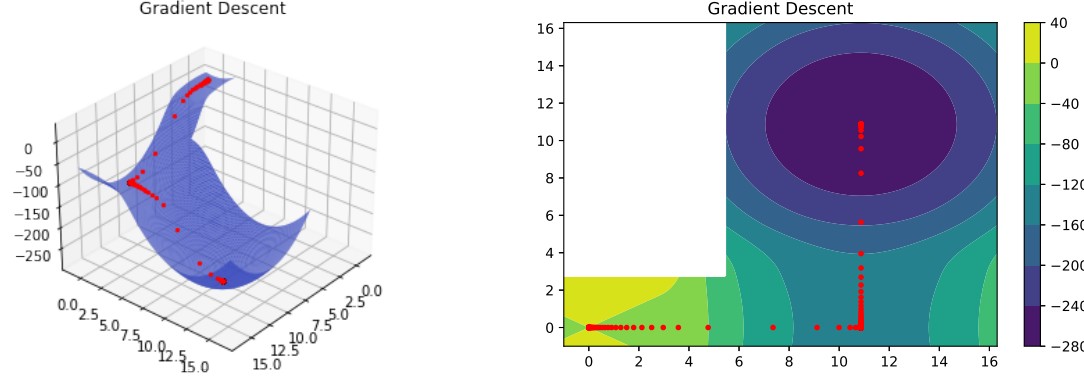

Figure 2: Surface plot and contour plot of a function $F : \mathbb{R}^2 \to \mathbb{R}$ defined in Du et al. (2017). We run gradient descent algorithm with a default step-size as suggested in Du et al. (2017) and initial point $x^{(0)} = (0.0001, 0.0001)^T$.

specified number of iterations (see Jin et al. (2017) for details). An alternative approach that is investigated now is to utilize GD-Kick (Algorithm 2).

In Figure 4 we compare the evolution of GD, and PGD with various perturbations[3] Note that the performance of PGD is sensitive on the choice of hyper-parameters. On the other hand, GD-Kick does not require tuning and we simply run it with $s = 2$; the performance is comparable to PGD.

We also used GD-Kick to minimize the quadratic model $m_k(\Delta) = F(x_k) + \langle \nabla F(x_k), \Delta \rangle + \frac{1}{2}\Delta^T \nabla^2 F(x_k)\Delta$ that fits the structure of equation 1 with $A = \nabla^2 F(x_k)$ and $b = -\nabla F(x_k)$. If negative curvature is discovered, we would go in that direction, otherwise we would take step towards the approximate solution $\bar{\Delta} \approx \min_\Delta m_k(\Delta)$ that was obtained by GD-EID. We denote this on Figure 4 as GD-EIG-NC. These numerical results support the use of GD-Kick to help escape saddle points.

---

[3]The noise that is added to the current iteration is sampled from sphere with radius $r$. The choice of $r = 2.718 \cdot 10^{-2}$ is a setting used by Du et al. (2017).

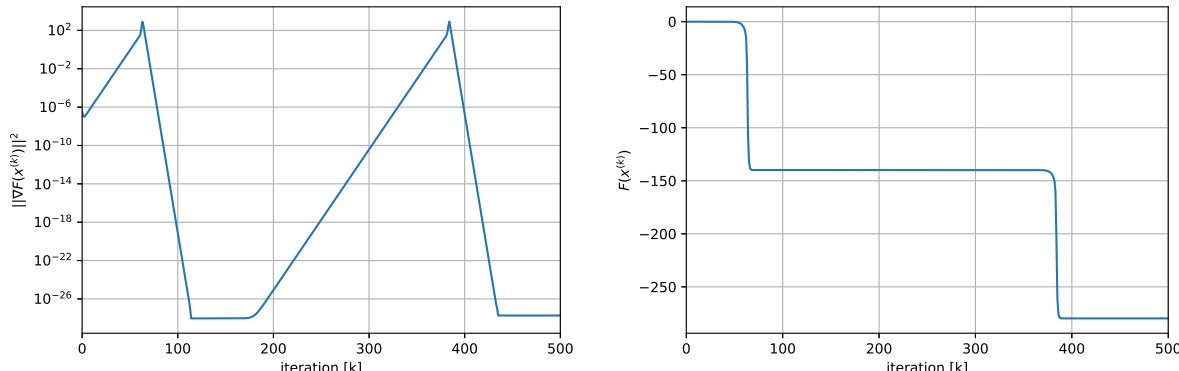

Figure 3: Evolution of $F(x^{(k)})$ and $\|\nabla F(x^{(k)})\|^2$ of iterates produced by GD algorithm.

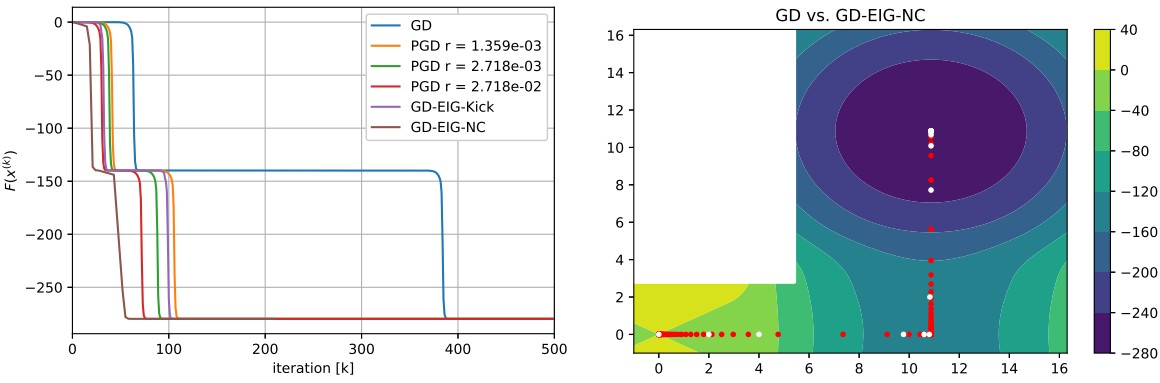

Figure 4: Left: Comparison of GD with PGD for various setting, together with GD-Kick. Note that for GD-Kick we count on x-axis the cost, not the iterations itself, the reason is that we need to evaluate multiple hessian-vector products when minimizing the model $m_k(\Delta)$. Right: Comparison of GD with GD-Kick. Note that the GD-Kick needs only a few iterations.

# 6 Conclusion

This work formalized the connection between GD with a fixed step size and the PM, both with and without fixed momentum, when applied to a quadratic function. Thus, GD implicitly provides an approximation to the leftmost eigen-pair of the Hessian. Several examples from recent literature show that GD with a fixed step size takes exponential time to escape saddle points. These examples were re-visited, to show that if 'freely available' eigeninformation was used, the performance of GD with an adaptive step size may be better than previously suggested. In particular, it may be possible to use an adaptive step length based on the estimate of the leftmost eigenvalue to escape saddle points more quickly than if a fixed step size is used.

A new algorithm called GD-Kick was presented, which uses the approximate eigenvalue information to attempt a long step, a 'kick', every $s$ iterations. GD-Kick is guaranteed to converge when $A$ is PD, and it behaves better in practice than vanilla GD. Numerical experiments confirmed that GD does provide an estimate of the leftmost eigen-pair in practice, although many iterations are needed for an accurate approximation. Examples were also presented that show the benefits of GD-Kick compared with vanilla GD, which supports the view that approximate eigen (curvature) information should be used within GD algorithms where possible.

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

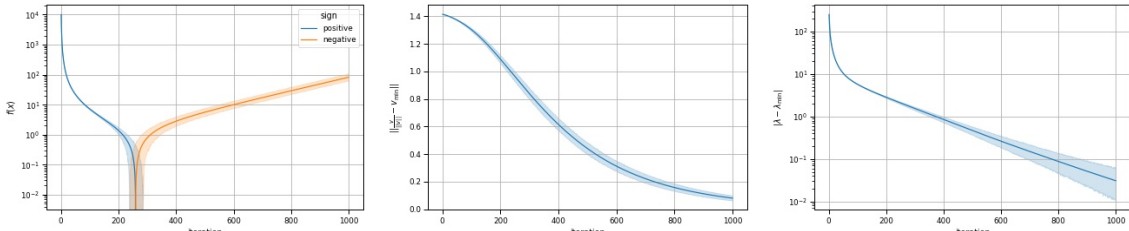

Figure 5: Figure showing the results of the first numerical experiment. The left plot shows the evolution of the function values, the middle plot shows the accuracy of the eigenvector approximation, and the right plot shows the accuracy of the eigenvalue estimate.

# A    Further numerical experiments

## A.1    Estimating the left eigen-pair

The purpose of these numerical experiments is to confirm, numerically, that an estimate of the leftmost eigen-pair can be obtained in practice using GD-Kick.

Here, 100 random symmetric indefinite matrices $A \in \mathbb{R}^{200 \times 200}$ were generated, and in all cases $b = 0$, so that $f(x) = \frac{1}{2} x^T A x$. For each matrix, a random initial starting point $x^{(0)} \in \mathbb{R}^{200}$ was generated, and GD-EIG, with a step size of $\alpha = 1/\lambda_1$, was run for 1,000 iterations. Note that, because the matrix $A$ is indefinite, there is a saddle point, and the optimization problem equation 1 is unbounded from below. Figure 5 shows the results of this experiment.

The left plot in Figure 5 shows the evolution of $f(x^{(k)})$ as the algorithm progresses. Note that, because a log-scale is used, positive and negative function values are differentiated by colour (blue corresponds to $f(x^{(k)}) \geq 0$, while red corresponds to $f(x^{(k)}) < 0$). All 100 runs/problem instances are plotted in this figure, giving a shaded region, with the average over the 100 runs given by the darker line. This confirms that GD-EIG is solving the problem equation 1, as expected ($f(x^{(k)}) \to -\infty$ as $k \to \infty$). Notice that the function value decreases rapidly initially, but then slows down as it nears the saddle point. After just over 200 iterations on average, a direction of negative curvature is found, the function value becomes negative and the iterates slowly escape the saddle point.

In the middle plot we show the accuracy of the left-most eigenvector estimate, and the right plot shows the left-most eigenvalue estimate. Recall that the Rayleigh quotient provides a quadratically accurate approximation to the eigenvalue, while the approximation to the eigenvector is only linear, which corresponds with what is shown numerically (i.e., it takes more iterations to generate an accurate approximation to the leftmost eigenvector, compared with the corresponding eigenvalue). This confirms that an estimate of the leftmost eigen-pair is recovered numerically. Recall equation 13, which gives the approximation rate for the leftmost eigenvalue, which is the same when using GD-EIG with a fixed step size, or using the PM. That is, we do not claim a faster rate of convergence using GD-EIG, but we do show that an approximation to the leftmost eigen-pair is available.

### A.1.1    Approximation using GD-Kick

Another experiment is performed to verify that GD-Kick can also be used to estimate a left eigen-pair. The experimental set up the same as in Section A.1, although for 2000 iterations. Figure 6 shows the evolution of the function value for GD and GD-Kick with several choices of $s$. Again, this plot distinguishes between positive and negative function values by colour: blue corresponds to $f(x^{(k)}) \geq 0$ and red corresponds to $f(x^{(k)}) < 0$. In each plot, the dashed line corresponds to GD with a fixed step size of $\alpha = 1/L$. For this representative problem instance using GD, the function value decreases rapidly initially, but then the iterates become trapped by the saddle point and the function values decrease slowly, until at approximately 1100 iterations, a direction of negative curvature is found, the function value switches to negative, and the iterates slowly escape the saddle point.

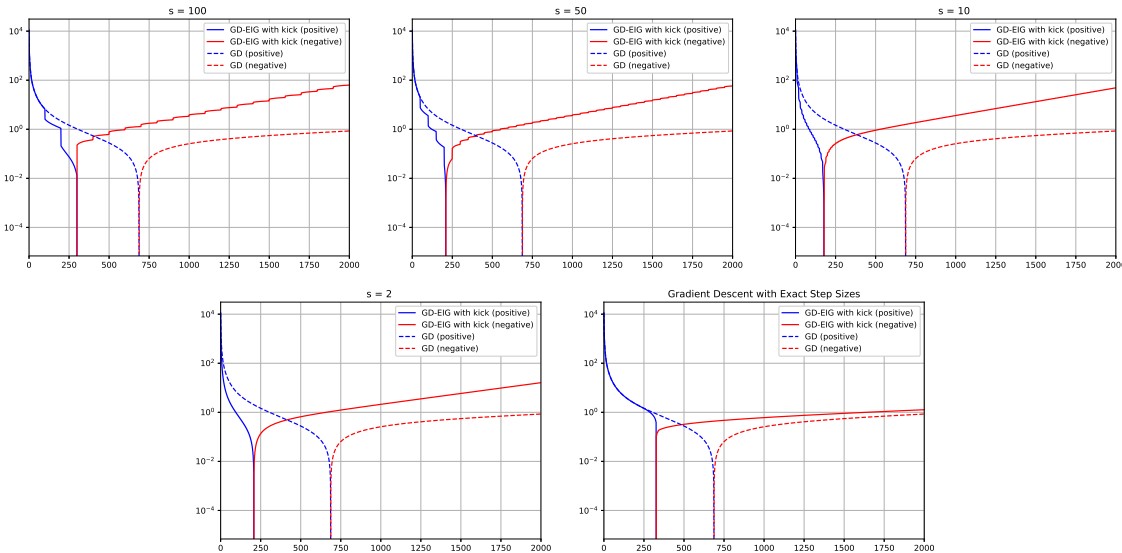

Figure 6: GD-EIG with kick.

Also shown in each plot is the behaviour of GD-Kick (solid line) with the stated $s$ value. In all cases, GD-Kick finds a direction of negative curvature more quickly than GD, and thus, escape from the saddle point is more rapid (the slope of the red (solid) curve is slightly steeper than for GD (dashed line) in each case). This confirms that using GD-Kick can be beneficial in practice. It also shows that the choice of $s$ (the number of iterations before a kick), impacts the practical performance of the algorithm. For this experiment, the best performance occurred when $s = 10$ (i.e., 9 iterations with a fixed step size, and then one kick step). It is also apparent that the 'kick' iterations lead to a larger reduction in the function value (the 'staircase' patterns). The case $s = 1$ is equivalent to Gradient/Steepest Descent with an exact step size and is included as a benchmark. Notice that, as expected, using an exact step size is better than GD with a fixed step size, but GD-Kick performs better in all cases.

## A.2   Investigation of step sizes in GD

This experiment investigates the behaviour of GD with the longer step size $\alpha = 2/L$. A symmetric positive definite matrix $A \in \mathbb{R}^{1000 \times 1000}$ was generated, as well as the optimal solution $x^* \in \mathbb{R}^{1000}$, with $b$ computed as $b = Ax^*$. A point $x^{(-1)} \in \mathbb{R}^{1000}$ was generated, and the starting point for the experiments was set as $x^{(0)} = x^{(-1)} - (1/\lambda_1)(Ax^{(-1)} - b)$. Figure 7 shows the results. Note that, while 100 problem instances of the type described above were generated, the results of a single, representative, problem instance are reported.

The lines correspond to the step sizes $\alpha = 1/L$ (blue), $\alpha = 2/L$ (red), and $\alpha = 2/(L + \mu)$ (green). The solid lines correspond to $\beta = 0$ (no momentum), the dashed lines correspond to $\beta = 0.5$, and the dotted lines correspond to $\beta = 0.8$. Notice that, regardless of the choice of momentum parameter $\beta$, the red and green lines are similar, and both choices led to improvements over the step size $\alpha = 1/L$. This is important because, while $L$ is often available when the algorithm begins, $\mu$ is typically unknown, so the step size $\alpha = 2/L$ is convenient in practice. These experiments also show that momentum can be helpful in terms of the practical behaviour of the method, with these experiments suggesting larger values of the momentum parameter correspond to a reduction in the number of iterations.

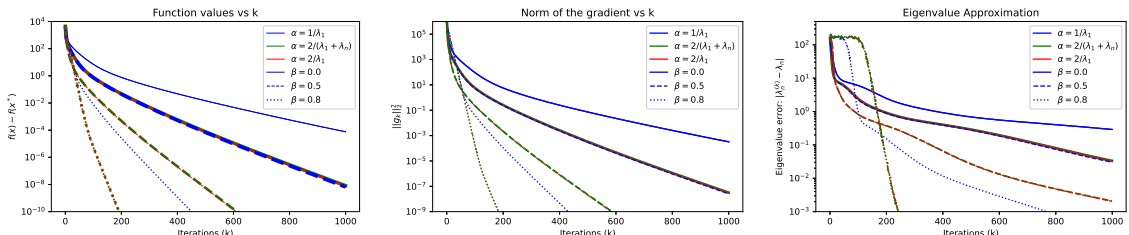

Figure 7: Investigating the choice of step length in Gradient Descent.

