# OpenReview forum: "Gradient Descent and the Power Method: Exploiting their connection to find the leftmost eigen-pair and escape saddle points"
_TMLR — Rejected by TMLR_

### Review · Reviewer_oubp · 2024-10-24

**Summary Of Contributions:**

The paper establishes a formal connection between Gradient Descent with Momentum (GDM) and the Power Method with Momentum (PMM) in the very specific context of unconstrained quadratic optimization problems. The main result is to show that running GDM on min_x 1/2<x,Ax> - <b,x> with fixed step size implicitly executes PMM on the gradients Ax-b, providing approximations to the leftmost eigenvalue-eigenvector pair of the Hessian matrix A. Based on this insight, they propose Gradient Descent with a Kick (GD-Kick), an algorithm that occasionally takes longer "kick" steps using the estimated eigenvalue information obtained through the GDM-PMM relationship. The paper provides theoretical convergence guarantees for GD-Kick in Theorem 1 and demonstrates through numerical experiments that it can outperform vanilla gradient descent on unconstrained quadratic problems, particularly near saddle points. The paper also revisits known examples from the literature where gradient descent exhibits slow escape from saddle points, showing how the estimated eigenvalue information can improve performance in these cases.

**Audience:**

Yes

**Broader Impact Concerns:**

The paper does not raise any ethical concerns given its abstract theoretical nature and focus on mathematical elements of the theory of optimization rather than concrete applications.

**Claims And Evidence:**

Yes

**Requested Changes:**

- The paper could acknowledge more clearly that the GDM-PMM connection is only relevant for unconstrained quadratic optimization problems, for instance by including “for quadratic problems” or something descriptive in the title.
- A numerical comparison with more standard methods for unconstrained quadratic optimization. A numerical investigation into GD-Kick for functions that are not quadratic.
- A discussion of how this analysis might or might not be extended or be useful for nonquadratic problems solved using gradient descent.
- Provide more discussion on choosing the parameter s (the frequency of the kick step) and its effect. The convergence results in Theorem 1 seem to not depend on s at all which is interesting and some discussion of this would benefit the clarity of the paper.
- A motivating application, for instance those described within the paper using GD-Kick as a subroutine in some broader optimization algorithm, would be greatly appreciated to demonstrate that the algorithm provides benefits for solving this class of problem vs other possible choices.

**Strengths And Weaknesses:**

Strengths
- The mathematical analysis is rigorous and technically sound.
- The connection between GDM and PMM, when applied to unconstrained quadratic minimization, appears to be novel.
- The arguments in the proofs are clearly written and the numerical experiments conform to the theoretical assumptions of the problem being studied.

Weaknesses
- The scope is narrow, focusing only on unconstrained quadratic problems which are typically solved using more effective methods than gradient descent (e.g., conjugate gradient methods, quasi-newton methods, MINRES, etc), even in high dimensional spaces. Although Theorem 1 is stated without assuming that the objective function f is quadratic, GD-Kick is no longer a first-order algorithm at that point. Indeed, it will involve the Hessian of the function f and this drawback is not discussed in detail. It's well-known that you can use the information gained by computing the Hessian to escape saddle points; the problem is that the Hessian is expensive to compute and/or store. So when it is claimed that GD-Kick only costs a couple of additional inner products, I find it misleading - this savings in cost is only happening because in the very specific case of f(x) = 1/2<x,Ax> - <b,x> the Hessian is just A. In the general case, you will have to compute (and store!) the Hessian of your objective function f and all the potential headaches that follow.
- The insights gained from this restricted setting do not generalize to nonquadratic problems nor to algorithm pairs besides GDM and PMM, nor is any study of this kind made in the paper.
- The numerical experiments do not compare against the state of the art methods for solving unconstrained quadratic problems. To contextualize the performance of this algorithm and the theoretical results, it is necessary to compare GD-Kick not just to GDM but also to other algorithms capable of solving unconstrained quadratic problems efficiently.
- I understand that on some level the point of the numerical experiments are to show that GD-Kick overcomes the saddle-point obstacles observed to plague ordinary GDM as pointed out in Du et al. 2017, Paternain et al. 2019 but I find the focus here is too concentrated on the specific examples used by Du et al. 2017, Paternain et al. 2019 rather than the overall point they were making, which is that GDM can get stuck even on simple functions. That we can cook up schemes to get out of saddle points when solving unconstrained quadratic optimization problems is known, as they are relatively simple objective functions - the pertinent question is whether something like this can be avoided/fixed in general (beyond quadratics) or at least in some relevant, broad class of functions. The fact that Theorem 1 applies to functions beyond quadratics is a good step in this direction but there is no corresponding numerical complement on nonquadratics in the paper.
- The discussion around the parameter s and its relevance to the convergence/stability of the algorithm is not well-developed.

---

> ### Author Response · Authors · 2025-02-27
> **Reviewer oubp: Requested Changes**
>
> Many thanks to reviewer oubp for taking the time to review this work and for the helpful feedback.
>
> We would like to start with a general comment about this work, because the reviewers have mentioned that the scope is ‘narrow’. The focus of this work is on two algorithms: Gradient Descent with Momentum (GDM), and the Power Method with Momentum (PMM). The PMM is used to find the dominant eigenvalue of a fixed matrix. This forces us to focus only on quadratic functions, because these are the only functions for which the Hessian is fixed. For any other function, the Hessian can change, and this immediately destroys any potential connection with the PMM (which must be applied to a single/fixed matrix). i.e., it is not that we are choosing to restrict the focus to quadratic functions, it is the PMM that forces the restriction to quadratic functions only. This is equivalent to the CG/MINRES–Lanczos connection, which holds precisely because CG/MINRES are for quadratic functions/systems of linear equations, and does not make sense in any other context because Lanczos finds the eigenvalues of a fixed matrix. We hope that this helps to clarify the necessity of specifically considering quadratic functions.
>
> We now comment on the "Requested Changes:"
>
> "The paper could acknowledge more clearly that the GDM-PMM connection is only relevant for unconstrained quadratic optimization problems, for instance by including “for quadratic problems” or something descriptive in the title."
> Thanks for the suggestion. We have changed the title to 'Gradient Descent and the Power Method: Exploiting their connection on quadratic functions to find the leftmost eigen-pair and escape saddle points'
>
> "A numerical comparison with more standard methods for unconstrained quadratic optimization. A numerical investigation into GD-Kick for functions that are not quadratic."
> Thanks for the comment. In Section 5.2, the objective function is not quadratic, it is a smooth nonconvex function. This is an important test-function, developed by Du et al 2017, to be representative of smooth machine learning problems/loss landscapes that are 'difficult for algorithms to navigate', because of the saddle points. (This test function essentially forces the iterates to encounter the saddle point before it can get to the local minimizer.) This numerical experiment shows that GD-Kick is successful on this smooth nonconvex optimization problem, and performs favourably with GD and Perturbed GD.
>
> "A discussion of how this analysis might or might not be extended or be useful for nonquadratic problems solved using gradient descent."
> Thanks for the suggestion. We have added a new paragraph to the introduction, summarizing that PM like methods require a fixed matrix, and that that is what necessitates study of quadratic functions.
>
> "Provide more discussion on choosing the parameter s (the frequency of the kick step) and its effect."
> Thanks for the suggestion. We do take your point that the parameter s impacts the performance of GD-Kick, and we are happy to provide some more numerical evidence for how this parameter might be chosen. In particular, we have run additional numerical experiments on a strongly convex quadratic function and compared the behaviour of GD-Kick with $s \in \{2,10,20,50,100,1000\}$, both with and without momentum. (The case $s=1000$ is essentially 999 iterations of vanilla GD, with the final iterate involving a 'kick' step, so it acts as a benchmark.) The numerical experiment show that an s value of 20, 50, 100 consistently performs well, and that overall s=20 is a good choice. Some intuition for why this is sensible is that the algorithm needs some time (iterates) to build up local curvature information about the function so that the 'kick' step is 'long enough'. Running the algorithm for further iterations will increase the accuracy of the approximation to the leftmost eigenvalue, but is unlikely to greatly impact the length of the 'kick' step. Thus, s=20 provides a nice middle ground for generating a kick step that is sufficiently long, but does not waste too many iterations ensuring the leftmost eigenvalue estimate is particularly accurate. If s is bigger than this then the kick step is taken too infrequently to make good progress, and if s is less than this the eigenvalue approximation is poor so that the algorithm doesn't reduce the function value enough. Moreover, these findings seem very consistent with the well known performance of PM like methods for generating eigenvalue approximations. We have included this as a new subsection (subsection 5.1) and provided similar discussion to what is written here.

---

> ### Author Response · Authors · 2025-02-27
> **Response continued**
>
> "The convergence results in Theorem 1 seem to not depend on s at all which is interesting and some discussion of this would benefit the clarity of the paper."
> Thanks for this comment. You're right that the result in Theorem 1 does not depend on s. The reason for this is that for a first order method  applied to a strongly convex problem, we can only ever expect to get linear convergence (and we have shown that indeed, the convergence is linear here). This is not an 'accelerated method', so we cannot expect the optimal rate. For general smooth strongly convex functions, GD-Kick involves a safeguard, (see Step 9 of Algorithm 2), so the kick-step is only accepted if it results in a function value no worse than vanilla GD (with a step length 1/L). Thus, GD-Kick inherits the convergence rate of vanilla GD. The question is "could this be improved?". We believe the answer is no, and the reasoning is as follows. To improve the constants in the rate, one would have to quantify the improvement (in terms of the function value reduction) for a kick step. The form of the kick step is effectively the reciprocal of the Rayleigh quotient, and it is well known that for any $v\in R^n$, $\mu \leq \frac{v^TAv}{v^Tv} \leq L$. So even though in practice it is likely to be the case that  $\mu \approx \frac{(g^{(k)})^TAg^{(k)}}{\|g^{(k)}\|_2^2} $, to maintain the inequalities in the proof one can only show that it is guaranteed that the reduction is no worse than $1/L$. We hope that this helps, and we have provided further elaboration on this matter following Theorem 1.
>
> "A motivating application, for instance those described within the paper using GD-Kick as a subroutine in some broader optimization algorithm, would be greatly appreciated to demonstrate that the algorithm provides benefits for solving this class of problem vs other possible choices."
> Thanks for the suggestion. We are running this now and hope to have it ready shortly.
>
> The reviewer has also added some interesting comments in the "Weaknesses" section, which we would be grateful for the opportunity to respond to:
>
> "Although Theorem 1 is stated without assuming that the objective function f is quadratic, GD-Kick is no longer a first-order algorithm at that point. Indeed, it will involve the Hessian of the function f and this drawback is not discussed in detail. It's well-known that you can use the information gained by computing the Hessian to escape saddle points; the problem is that the Hessian is expensive to compute and/or store. So when it is claimed that GD-Kick only costs a couple of additional inner products, I find it misleading - this savings in cost is only happening because in the very specific case of f(x) = 1/2<x,Ax> - <b,x> the Hessian is just A. In the general case, you will have to compute (and store!) the Hessian of your objective function f and all the potential headaches that follow."
> Thanks for your thoughts on this. The main contribution of this work is establishing a formal connection between GDM when applied to quadratic functions and the PMM. The question then is, ‘how does this help?’, so we have suggested a way of adding in the local curvature/eigenvalue into a gradient based algorithm, GD- Kick. This algorithm seems sensible for quadratic functions (and the numerical experiments support this assertion). If the function is not quadratic, this algorithm is only intended as the most basic ‘prototype’ algorithm one could imagine, and we do not necessarily think one should use it in practice (although the numerical experiment in Section 5.2 shows that it can be successful on non-quadratic problems). Nevertheless, if one wanted to use GD-Kick for a non-quadratic function, the algorithm is simply GDM with a fixed step size 1/L and only every s iterations is a ‘kick step’ attempted. At that iteration in Step 5 of Algorithm 2 one must compute ν(k) = uT ∇f (x(k))/∥∇f (x(k))∥2^2, where u = ∇2f (x(k))∇f (x(k)), i.e., one matrix vector product with the Hessian is required every sth iteration of Algorithm 2. For some applications it may be possible to compute the matrix vector product ∇2f (x(k))∇f (x(k)) in a ‘matrix-free’ way (i.e., only using an oracle/black- box), and if s is ‘large’, then this may not be too much of a burden, and GD-Kick may still be a reasonable algorithm to use. Of course, the reviewer is correct that there may be some objective functions for which even these moderate computations are too expensive, in which case this may not be a suitable choice of algorithm. This trade-off is typical for large scale problems.

---

> ### Author Response · Authors · 2025-02-27
> **Response continued**
>
> "I understand that on some level the point of the numerical experiments are to show that GD-Kick overcomes the saddle-point obstacles observed to plague ordinary GDM as pointed out in Du et al. 2017, Paternain et al. 2019 but I find the focus here is too concentrated on the specific examples used by Du et al. 2017, Paternain et al. 2019 rather than the overall point they were making, which is that GDM can get stuck even on simple functions. That we can cook up schemes to get out of saddle points when solving unconstrained quadratic optimization problems is known, as they are relatively simple objective functions - the pertinent question is whether something like this can be avoided/fixed in general (beyond quadratics) or at least in some relevant, broad class of functions. The fact that Theorem 1 applies to functions beyond quadratics is a good step in this direction but there is no corresponding numerical complement on nonquadratics in the paper."
> Thank you for this comment, which is really insightful. We fully agree that the crux of their argument is that GDM can get stuck on simple quadratic functions. Our reply is that the overall point we hoped to make is the question ‘but does GDM really get stuck, or is it being underestimated as an algorithm?’. This is why we feel that the major contribution of this work is establishing the GDM–PMM connection. It shows that when GDM is applied to quadratic functions, eigeninformation is available, so is it really fair to say the algorithm is getting ‘stuck’, or is it that we have not previously been using it properly (we haven’t been exploiting the freely available eigen-information that can help the algorithm navigate saddle points)? The research field has strongly taken up the perspective of Du et al. 2017 and Paternain et al.
> 2019 that GDM is no good for saddle points, and this has led to all the (very good) research into alternative methods for nonconvex functions. We are simply trying to say that, given that we establish a formal relationship between GDM and the PMM, perhaps gradient based methods should not be written off.
>
> Thanks again, and we are very grateful for the helpful feedback and interesting questions.

---

> > ### Comment · Reviewer_oubp · 2025-03-02
> >
> > Thank you for the reponse to my feedback. My issues and questions about the paper have all been adequately addressed now.

---

### Review · Reviewer_Ftsj · 2024-10-30

**Summary Of Contributions:**

This paper examines the relationship between Gradient Descent with fixed Momentum (GDM) and the Power Method with fixed Momentum (PMM): under certain conditions, GDM with a fixed step size implicitly performs the Power Method on gradients, thereby computing the dominant eigen-pair of the Hessian. Leveraging this relationship, the authors propose a method, Gradient Descent with a Kick (GD-Kick), which attempts to speed up gradient descent by periodically adjusting step sizes based on curvature. They also validate GD-Kick with numerical experiments, showcasing its advantages over standard GD, especially in scenarios where GD struggles near saddle points.

**Audience:**

Yes

**Claims And Evidence:**

Yes

**Requested Changes:**

1. As mentioned in weaknesses, please consider to add

       (1) discussion of limitation and potential adaptation to non-quadratic non-convex problems.

       (2) add introduction and comparison to related works on saddle-point-escaping techniques in GD.

2. In figure 1 center column, could you explain why the large step-size is more frequent when $s=100$? Wouldn't the large step-size be more frequent for smaller $s$?

3. On page 14, it is mentioned GD-kick doesn't require tuning. But there is a parameter, $s$, to tune in the algorithm. At least, from the experiments shown in the paper, the choice of s does affect the performance significantly. For example, in figure 1 first row (PD matrix), $s=20$ outperforms $s=100$ obviously. However, in figure 1 second row (semi-PD matrix), $s=20$ and $s=100$ have comparable performance. Could you analytically/empirically explain how to pick this parameter $s$? Would a smaller $s$ always be preferable? Does the choice of s depend on the matrix $A$?

4. Please check the typos in the paper. For example,

       (1) page 7: Section **??**

       (2) page 9: The eigen-decomposition **is** $H$ in equation $(5)$ is ...

       (3) page 13: This provided **motivated** to investigate ...

**Strengths And Weaknesses:**

**Strengths**

The paper addresses a challenge in nonconvex optimization—escaping saddle points. By enhancing GD with adaptive steps using curvature information which could be obtained at low cost , the proposed GD-Kick method has the potential to benefit a wider range of applications in machine learning and optimization.

**Weaknesses**

1. The current work primarily focuses on quadratic functions. It would be beneficial to discuss the potential limitations or adaptations required for GD-Kick when applied to more general non-quadratic, nonconvex problems.

2. Although GD-Kick shows promising results, it would be helpful to compare to other established saddle-point-escaping techniques ( other than Perturbed Gradient Descent). These techniques should also be mentioned in the related works.

---

### Review · Reviewer_FDbH · 2025-02-15

**Summary Of Contributions:**

The paper establishes the connection between Gradient Descent with   Momentum (GDM) and the power method for nonconvex quadratic functions. The authors introduce a new algorithm called Gradient Descent with a Kick (GD-Kick). It shows that GDM implicitly helps approximate the leftmost eigenvalue of the Hessian, thereby enhancing convergence near saddle points. The authors suggest that leveraging the eigenvalue information can lead to a faster escape from saddle points by using adaptive step sizes. GD-Kick performs better in numerical experiments than standard Gradient Descent, especially when dealing with saddle points.

**Audience:**

Yes

**Claims And Evidence:**

Yes

**Requested Changes:**

see the  Weaknesses part.

**Strengths And Weaknesses:**

Strengths: 1. The paper establishes a significant connection between GDM and the power method, demonstrating that applying GDM to nonconvex quadratic functions effectively leverages eigen-information.

2. The authors revisit known challenges of escaping saddle points using GD and propose utilizing eigenvalue approximations to enhance the algorithm's capacity to escape these critical points more efficiently.

3. The introduction of GD-Kick is an inventive strategy that combines traditional gradient descent with adaptive longer steps based on curvature information.

Weaknesses: 1. As a machine learning journal, the authors should provide more detailed information on nonconvex quadratic optimization within the field of machine learning. Specifically, they should discuss its applications and the associated challenges.

2. While the authors offer some theoretical explanations for the method, the current theory is insufficient to substantiate the claims regarding the advantages of the new algorithm.

3. The numerical results are insufficient. Additional tests in the field of machine learning are necessary.

---

### Decision · Action_Editor_DgX6 · 2025-04-30

**Recommendation:** Reject

**Comment:**

As mentioned above, the reviewers are in agreement about the suitability of this work to TMLR. On the other hand, the authors did not submit a rebuttal to address the concerns of the reviewers Ftsj and FDbH. On the other hand, the authors submitted a very detailed rebuttal to Reviewer oubp, which in my reading, addresses some concerns raised by all the reviewers. However, the authors are recommended to respond to all the reviewers and directly respond to their concerns (or explain to the AE why they have not so) and update their paper accordingly to address the concerns and then resubmit. We will then try to assign the paper to the same reviewers.

**Audience:**

The reviewing team agrees that the findings of this paper are of interest to the audience of TMLR.

**Claims And Evidence:**

The reviewing team has concerns on this front. Unfortunately, these concerns do not seem to have been addressed by the author rebuttal. In particular, while the reviewers agree that the connection made in this work about PMM and GDM and other conclusions of the work are interesting, they asked questions regarding the difficulties of extending to more general problem classes relevant to ML as well as more in-depth comparisons about the related work. The authors' rebuttals were missing for the reviewers Ftsj and FDbH. Even though part of the author rebuttal to Reviewer oubp has points that address some of the concerns of the other two reviewers (including the extensions beyond quadratic functions, etc.), the authors are expected to respond to all the reviewers and address their concerns directly.

**Resubmission Of Major Revision:**

The authors may consider submitting a major revision at a later time.